# Characterising the loss-of-function impact of 5′ untranslated region variants in 15,708 individuals

Nicola Whiffin [1,2,3], Konrad J. Karczewski [3,4], Xiaolei Zhang [1,2], Sonia Chothani[5], Miriam J. Smith[6], D. Gareth Evans[6], Angharad M. Roberts[1,2], Nicholas M. Quaife [1,2], Sebastian Schafer [5,7], Owen Rackham[5], Jessica Alföldi [3,4], Anne H. O'Donnell-Luria [3,8,9], Laurent C. Francioli [3,4], Genome Aggregation Database Production Team, Genome Aggregation Database Consortium, Stuart A. Cook [1,5,7], Paul J.R. Barton [1,2], Daniel G. MacArthur [3,4,10,11,152] & James S. Ware [1,2,3,152]

Upstream open reading frames (uORFs) are tissue-specific *cis*-regulators of protein translation. Isolated reports have shown that variants that create or disrupt uORFs can cause disease. Here, in a systematic genome-wide study using 15,708 whole genome sequences, we show that variants that create new upstream start codons, and variants disrupting stop sites of existing uORFs, are under strong negative selection. This selection signal is significantly stronger for variants arising upstream of genes intolerant to loss-of-function variants. Furthermore, variants creating uORFs that overlap the coding sequence show signals of selection equivalent to coding missense variants. Finally, we identify specific genes where modification of uORFs likely represents an important disease mechanism, and report a novel uORF frameshift variant upstream of *NF2* in neurofibromatosis. Our results highlight uORF-perturbing variants as an under-recognised functional class that contribute to penetrant human disease, and demonstrate the power of large-scale population sequencing data in studying non-coding variant classes.

[1] National Heart and Lung Institute and MRC London Institute of Medical Sciences, Imperial College London, Du Cane Road, London W12 0NN, UK. [2] NIHR Royal Brompton Cardiovascular Research Centre, Royal Brompton and Harefield National Health Service Foundation Trust, Sydney Street, London SW3 6NP, UK. [3] Medical and Population Genetics, Broad Institute of MIT and Harvard, 415 Main Street, Cambridge, MA 02142, USA. [4] Analytical and Translational Genetics Unit, Massachusetts General Hospital, 55 Fruit Street, Boston, MA 02114, USA. [5] Program in Cardiovascular and Metabolic Disorders, Duke-NUS Medical School, 8 College Road, Singapore 169857, Singapore. [6] NW Genomic Laboratory Hub, Centre for Genomic Medicine, Division of Evolution and Genomic Science, St Mary's Hospital, University of Manchester, Oxford Road, Manchester M13 9WL, UK. [7] National Heart Centre Singapore, 5 Hospital Drive, Singapore 169609, Singapore. [8] Division of Genetics and Genomics, Boston Children's Hospital, Boston, MA 02115, USA. [9] Department of Pediatrics, Harvard Medical School, Boston, MA 02115, USA. [10] Centre for Population Genomics, Garvan Institute of Medical Research, and UNSW Sydney, Sydney, Australia. [11] Centre for Population Genomics, Murdoch Children's Research Institute, Melbourne, Australia. [152] These authors contributed equally: Daniel G. MacArthur, James S. Ware. A full list of consortium members appears at the end of the paper. Correspondence and requests for materials should be addressed to N.W. (email: n.whiffin@imperial.ac.uk)

Upstream open reading frames (uORFs) are ORFs encoded within the 5' untranslated regions (5'UTRs) of protein coding genes. uORFs are found upstream of around half of all known genes[1], and are important tissue-specific *cis*-regulators of translation. Active translation of a uORF typically reduces downstream protein levels by up to 80%[1]. There are strong signatures of negative selection acting on these elements, with fewer upstream start codons (uAUGs) present in the human genome than would be expected by chance[1–3]. In addition, the start codons of uORFs have been shown to be the most conserved sites in 5'UTRs[1], supporting the importance of uORFs in the regulation of protein levels.

In humans, translation is initiated when the small ribosomal subunit, which scans from the 5' end of the mRNA, recognises an AUG start codon[4]. The likelihood of an AUG initiating translation is dependent on local sequence context, and in particular the degree of similarity to the Kozak consensus sequence[5,6]. uORFs can inhibit translation through multiple mechanisms. For some genes, uORFs may be translated into a small peptide which can directly inhibit translation by interacting with and stalling the elongating ribosome at or near the uORF stop codon, creating a 'roadblock' for other scanning ribosomes[7,8]. It is also possible for this small peptide to have a distinct biological function[9]; however, in general uORFs do not show strong evidence for conservation of their amino acid sequence[2,10]. For other genes, translation from a uAUG appears to be sufficient to inhibit translation of the downstream protein, with the small uORF peptide only produced as a by-product.

Mechanisms of leaky scanning (whereby a scanning ribosome may bypass an uAUG), re-initiation (where the small ribosomal subunit remains bound to the mRNA and translation is re-initiated at the canonical AUG), and the existence of internal ribosome entry sites (from which the ribosome can start scanning part-way along the RNA), can all act to attenuate inhibition by uORFs, adding to the complexity of translational regulation[10–12]. Termination at a uORF stop codon can also trigger the nonsense-mediated decay pathway, further magnifying the inhibitory effects of uORFs[11,13]. To date, studies of translational regulation by individual uORFs have mainly been restricted to model organisms.

Recently, large scale studies have assessed the global translational repression ability of uORFs: in vertebrates, uORF-containing transcripts are globally less efficiently translated than mRNAs lacking uORFs, with this effect mediated by features of both sequence and structure[2]. Similarly, polysome profiling of 300,000 synthetic 5'UTRs identified uORFs and uAUGs as strongly repressive of translation, with the strength of repression dependent on the surrounding Kozak consensus sequence[14].

Although 5'UTRs are typically not assessed for variation in either clinical or research settings, having been excluded from most exome capture target regions, there are several documented examples of variants that create or disrupt a uORF playing a role in human disease[1,15–21]. These studies have focused on single gene disorders or candidate gene lists, often when no causal variant was identified in the coding sequence. No study to date has characterised the baseline population incidence of these variants.

Here we describe a systematic genome-wide study of variants that create and disrupt human uORFs, and characterise the contribution of this class of variation to human genetic disease. We use the allele frequency spectrum of variants in 15,708 whole-genome sequenced individuals from the Genome Aggregation Database (gnomAD)[22] to explore selection against variants that either create uAUGs or remove the stop codon of existing uORFs. Finally, we demonstrate that these variants make an under-recognised contribution to genetic disease.

## Results

**uAUG-creating variants are under strong negative selection.** To estimate the deleteriousness of variants that create a novel AUG start codon upstream of the canonical coding sequence (CDS), we assessed the frequency spectrum of uAUG-creating variants observed in gnomAD (Fig. 1a). We identified all possible single nucleotide variants (SNVs) in the UTRs of 18,593 canonical gene transcripts (see Methods) that would create a new uAUG, yielding 562,196 possible SNVs, an average of 30.2 per gene (Fig. 1b). Of these, 15,239 (2.7%) were observed at least once in whole genome sequence data from 15,708 individuals in gnomAD (Supplementary Fig. 1a), upstream of 7697 distinct genes.

We compared the mutability adjusted proportion of singletons (MAPS) score, a measure of the strength of selection acting against a variant class[23], for 14,897 observed high-quality autosomal uAUG-creating SNVs to other classes of coding and non-coding SNVs (see methods). As negative selection acts to prevent deleterious variants from increasing in frequency, damaging classes of variants have skewed frequency spectra, with a higher proportion appearing as singletons (i.e., observed only once in the gnomAD data set),[23] reflected in a higher MAPS score. Whilst all observed UTR SNVs have an overall MAPS score almost identical to synonymous variants, uAUG-creating SNVs have a significantly higher MAPS score (permuted $P < 1 \times 10^{-4}$; Fig. 1c), indicating a considerable selective pressure acting to remove these from the population.

We next evaluated subsets of uAUG-creating variants predicted to have distinct functional consequences. In addition to creating distinct uORFs, uAUGs may result in overlapping ORFs (oORFs) where the absence of an in-frame stop codon within the UTR results in an ORF that reads into the coding sequence, either in-frame (elongating the CDS), or out-of-frame (Fig. 1a). uAUG-creating variants that form oORFs have a significantly higher MAPS score than uORF-creating variants (permuted $P < 1 \times 10^{-4}$), and equivalent to missense variants in coding regions (Fig. 1c; Supplementary Fig. 1a).

We also investigated the context of uAUG-creating variants and find that uAUGs created within 50 bp of the CDS have higher MAPS than those created further away (permuted $P = 0.0042$), although this may be driven by the higher propensity of these variants to form oORFs. We did not observe a significantly greater MAPS score for uAUG-creating variants arising on a background of a strong Kozak consensus, though we observe a trend in this direction (Fig. 1c).

Given that uAUGs are expected to dramatically decrease downstream protein levels, we hypothesised that uAUG-creating variants would behave similarly to pLoF variants and thus be more deleterious when arising upstream of genes intolerant to LoF variation. Indeed, we show a significantly higher MAPS score for uAUG-creating SNVs upstream of genes which are most intolerant to pLoF variants (top sextile of LOEUF score[22]; 3193 genes) when compared to those that are most tolerant (bottom sextile; permuted $P < 1 \times 10^{-4}$; Fig. 1c). To ensure that the observed increase in MAPS score upstream of pLoF intolerant genes is not purely because the UTRs of these genes are more highly conserved, we compared the conservation of potential uAUG sites with the remainder of the 5'UTR, across all sextiles of LOEUF score. Overall, a significantly higher proportion of possible uAUG-creating bases have phyloP scores >2 (10.3%), when compared to all other UTR bases (8.6%; Fisher's $P < 1 \times 10^{-100}$), with the size of this effect increasing as the corresponding genes become more intolerant to pLoF variants (Supplementary Fig. 2).

Next, we calculated MAPS for uAUG-creating variants arising upstream of 1,659 genes known to cause developmental disorders (DD; confirmed or probable genes from the Developmental

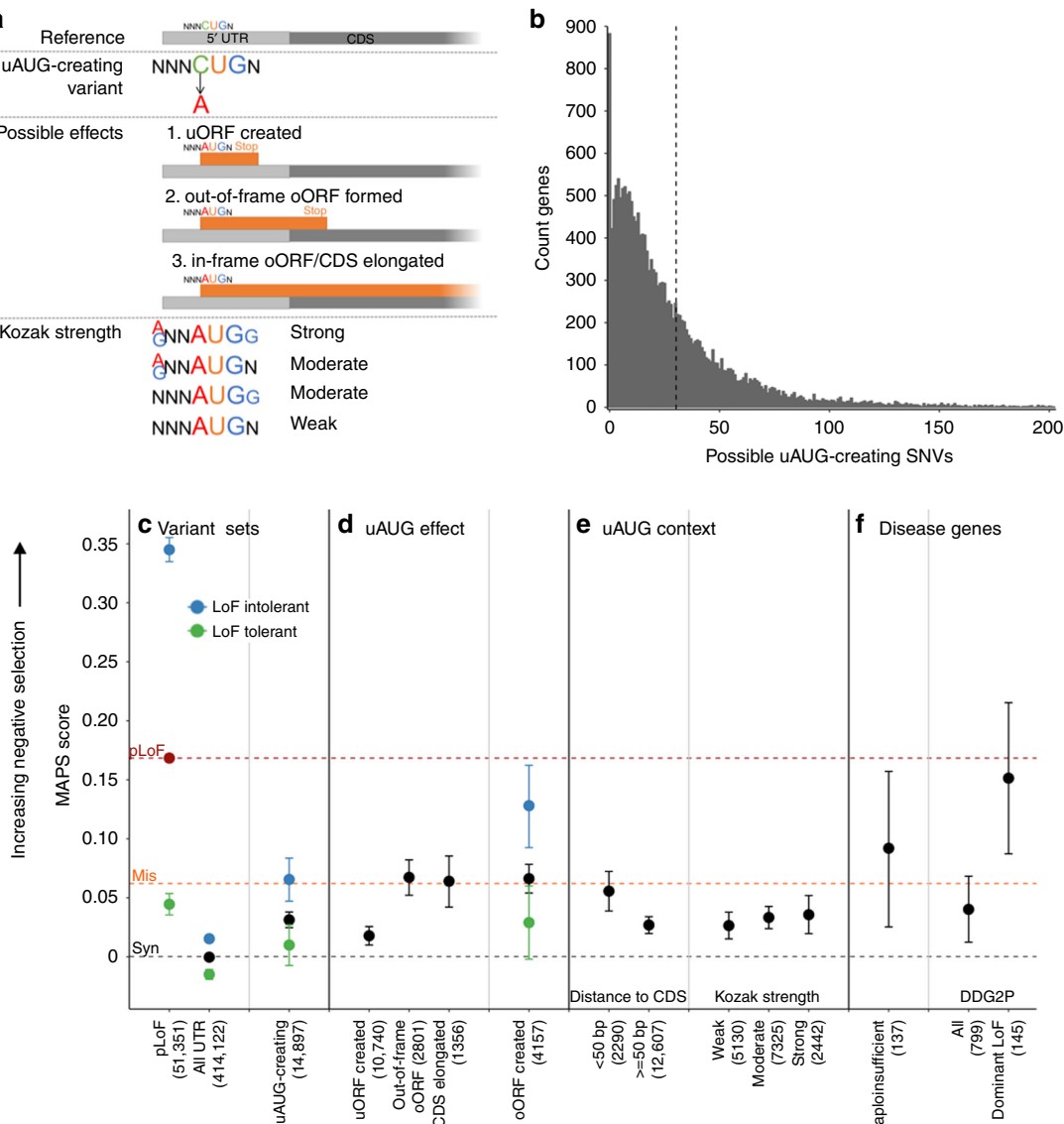

**Fig. 1** uAUG-creating variants have strong signals of negative selection, suggesting they are deleterious. **a** Schematic of uAUG-creating variants, their possible effects and how the strength of the surrounding Kozak consensus is determined. **b** The number of possible uAUG-creating SNVs in each of 18,593 genes, truncated at 200 (159 genes have >200). In total we identified 562,196 possible uAUG-creating SNVs, an average of 30.2 per gene (dotted line), with 883 genes having none. **c–f** MAPS scores (a measure of negative selection) for different variant sets. The number of observed variants for each set is shown in brackets. MAPS for classes of protein-coding SNVs are shown as dotted lines for comparison (synonymous–grey, missense–orange, and predicted loss-of-function (pLoF)–red point and red dotted line). Errors bars were calculated using bootstrapping (see methods). **c** While overall UTR variants display a selection signature similar to synonymous variants, uAUG-creating variants have significantly higher MAPS (indicative of being more deleterious; permuted $P < 1 \times 10^{-4}$). Variants are further subdivided into those upstream of, or within genes tolerant (green dot) and intolerant (blue dot) to LoF[22], with uAUG-creating variants upstream of LoF intolerant genes showing significantly stronger signals of selection than those upstream of LoF tolerant genes (permuted $P = 1 \times 10^{-4}$). pLoF variants are likewise stratified for comparison. **d** uAUG-creating variants that create an oORF or elongate the CDS show a significantly higher signal of selection than uORF-creating variants ($P < 1 \times 10^{-4}$; oORF created:out-of-frame oORF and CDS elongated combined). **e** The deleteriousness of uAUG-creating variants depends on the context into which they are created, with stronger selection against uAUG-creation close to the CDS, and with a stronger Kozak consensus sequence. **f** uAUG-creating variants are under strong negative selection upstream of genes manually curated as haploinsufficient[26] and developmental disorder genes reported to act via a dominant LoF mechanism. Abbreviations: CDS coding sequence, uAUG upstream AUG, uORF upstream open reading frame, oORF overlapping open reading frame, MAPS mutability adjusted proportion of singletons, pLoF predicted loss-of-function, DDG2P Developmental Disease Gene to Phenotype

Disease Gene to Phenotype (DDG2P) database). While uAUG-creating variants upstream of all DD genes do not show a signal of selection above all observed uAUG-creating variants, the MAPS score is significantly inflated when limiting to 279 DD genes with a known dominant LoF mechanism (permuted $P = 0.0012$; Fig. 1c).

**Variants that disrupt uORF stop codons are selected against.** As uAUG-creating variants that form oORFs have a significantly higher MAPS score than those with an in-frame UTR stop codon, we hypothesised that variants that disrupt the stop site of existing uORFs should also be under selection (Fig. 2a). These stop-removing variants could either be SNVs that change the

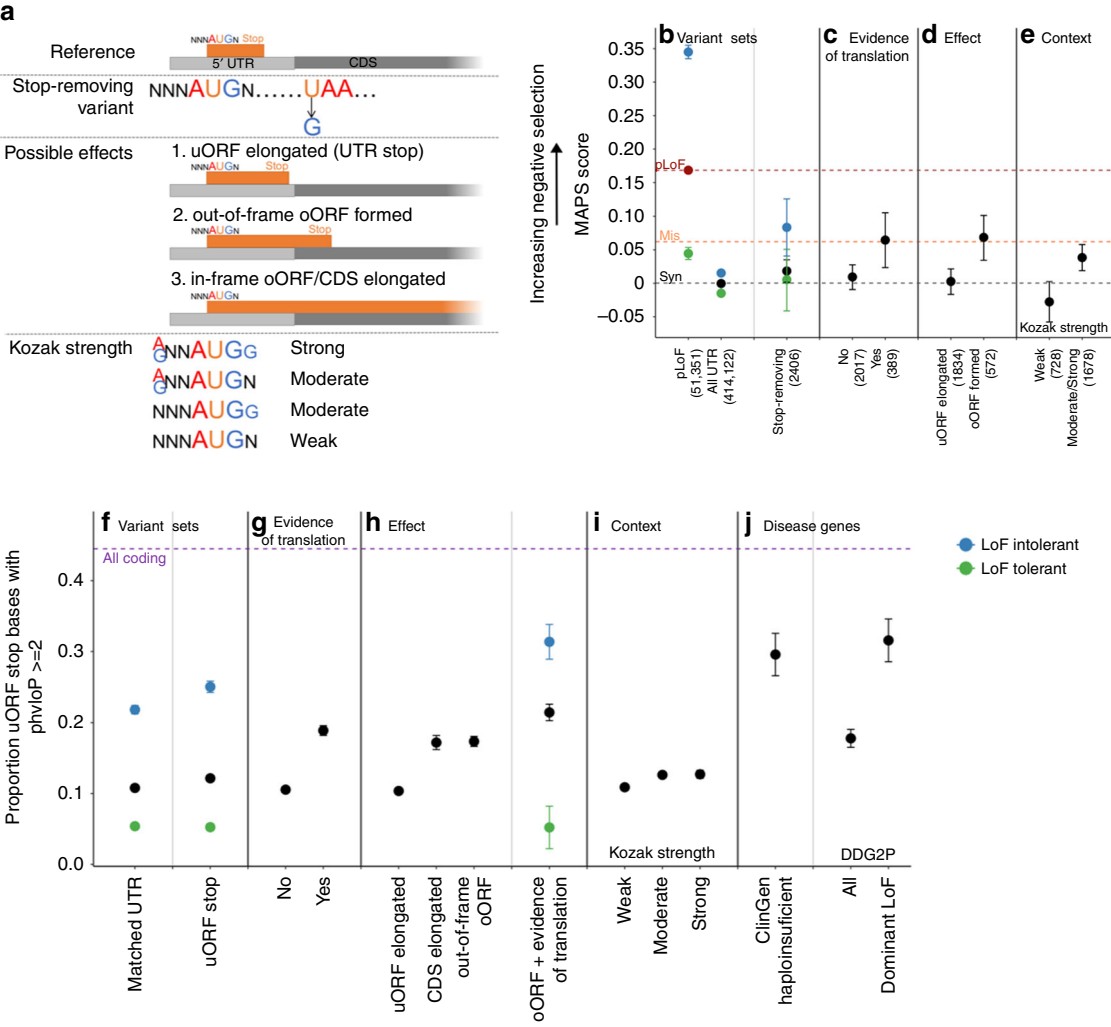

**Fig. 2** uORF stop codons are highly conserved and stop-removing variants show strong signals of negative selection. **a** Schematic of uORF stop-removing variants, their possible effects, and how the strength of the surrounding Kozak consensus is determined. **b–e** MAPS scores (a measure of negative selection) for different variant sets. The number of observed variants for each set is shown in brackets. MAPS for classes of protein-coding SNVs are shown as dotted lines for comparison (synonymous–black, missense–orange and predicted loss-of-function (pLoF)–red point and red dotted line). Confidence intervals were calculated using bootstrapping (see methods). **b** Stop-removing SNVs have a nominally higher MAPS score than all UTR SNVs (permuted $P = 0.030$). Variants are further subdivided into those upstream of, or within genes tolerant (green dot) and intolerant (blue dot) to LoF[22], with pLoF variants likewise stratified for comparison. Stop-removing SNVs (**c**) with evidence of translation (in sorfs.org) and (**d**) that create an oORF have signals of selection equivalent to missense variants. **e** A significantly higher MAPS is calculated for stop-removing variants where the uORF start site has a strong/moderate Kozak consensus, compared to those with a weak Kozak (permuted $P = 7 \times 10^{-4}$). **f–j** Since MAPS is only calculated on observed variants, we also looked at the conservation of all possible uORF stop site bases, reporting the proportion of bases with phyloP scores >2. All coding bases are shown as a purple dotted line for comparison. **f** The stop sites of predicted uORFs are significantly more conserved than all UTR bases matched on gene and distance from the CDS (Fisher's $P = 1.8 \times 10^{-17}$). uORF stop bases are most highly conserved when (**g**) the uORF has evidence of translation, (**h**) the variant results in an oORF, (**i**) the uORF start site has a strong/moderate Kozak consensus, and (**j**) upstream of curated haploinsufficient genes and developmental genes with a known dominant LoF disease mechanism. Error bars represent 95% binomial confidence intervals. CDS coding sequence, uORF upstream open reading frame, oORF overlapping open reading frame, MAPS mutability adjusted proportion of singletons, DDG2P Developmental Disease Gene to Phenotype

termination codon to one that codes for an amino acid, or frameshifting indels within the uORF sequence that cause the uORF to read through the normal stop codon. If there is no other in-frame stop codon before the CDS will result in an oORF.

We identified all possible SNVs that would remove the stop codon of a predicted uORF ($n = 169,206$; see methods), and calculated the MAPS score for 2,406 such variants observed in gnomAD. Stop-removing SNVs have a nominally higher MAPS score than all UTR SNVs (permuted $P = 0.030$). This difference is greater when specifically considering stop-removing SNVs which are upstream of LoF intolerant genes (permuted $P = 0.0012$),

result in an oORF (permuted $P = 2 \times 10^{-4}$), or where the uORF has either prior evidence of translation (documented in sorfs.org[24]; permuted $P = 0.0049$), or a strong/moderate Kozak consensus (permuted $P = 7 \times 10^{-4}$; Fig. 2b).

As the power of MAPS is limited by the small number of stop-removing variants in each category observed in gnomAD, we performed a complementary analysis investigating base level conservation at all uORF stop sites using PhyloP[25]. A significantly greater proportion of uORF stop site bases have PhyloP scores >2 (12.2%) compared to UTR bases matched by gene and distance from the CDS (10.8%; Fisher's $P = 1.8 \times 10^{-17}$; Fig. 2c). This

proportion is significantly higher where there is evidence supporting translation of the uORF (18.9%; Fisher's $P = 3.6 \times 10^{-83}$) or when removing the stop would result in an oORF (either in-frame or out-of-frame; 17.2% and 17.4%, respectively; Fisher's $P = 3.0 \times 10^{-25}$ and $2.6 \times 10^{-47}$, respectively). Furthermore, a greater proportion of stop site bases have PhyloP scores >2 when the uORF start codon has a strong or moderate Kozak when compared to a weak Kozak consensus (12.7% vs 10.9%; Fisher's $P = 5.5 \times 10^{-10}$; matched UTR bases Fisher's $P = 0.88$; Fig. 2c).

The increased power of this analysis enables us to convincingly demonstrate that uORF stop sites upstream of (1) LoF intolerant genes, (2) genes manually curated as haploinsufficient[26], and (3) developmental disorder genes with a dominant LoF mechanism, are all highly conserved. Stop sites upstream of genes in these groups have 21.9%, 29.6% and 31.6% of bases with PhyloP >2, respectively (Fisher's $P = 8.2 \times 10^{-250}$, $4.7 \times 10^{-43}$ and $1.4 \times 10^{-52}$ compared to all stop site bases, respectively; Fig. 2c), suggesting that removing these stop sites is likely to be deleterious.

**Specific genes are sensitive to uAUG-perturbing variants.** We searched the Human Gene Mutation Database (HGMD)[27] and ClinVar[28] for uORF-creating or uORF-disrupting variants, identifying 39 uAUG-creating and four stop-removing (likely) pathogenic/disease mutations in 37 different genes. All four stop-removing variants disrupt uORFs with uAUGs in a strong or moderate Kozak consensus and result in an oORF overlapping the CDS (Supplementary Table 2). Compared to all possible uAUG-creating variants in these 37 genes, the 39 reported disease-causing uAUG-creating variants (Supplementary Table 1) are significantly more likely to be created into a moderate or strong Kozak consensus (binomial $P = 3.5 \times 10^{-4}$), create an out-of-frame oORF (binomial $P = 1.1 \times 10^{-5}$), and be within 50 bp of the CDS (binomial $P = 3.9 \times 10^{-7}$; Fig. 3a). These results support the assertion that the variant classes identified by MAPS as under strongest negative selection are most likely to be disease causing.

This analysis highlights disease genes where aberrant translational regulation through uORFs is an important disease mechanism. Previous analysis of the *NF1* gene in 361 patients with neurofibromatosis type 1 identified four 5'UTR variants as putatively disease-causing[29]. These variants were found in six unrelated probands, all of whom were negative for coding variants in both *NF1* and *SPRED1*. Three of the four variants either occurred de novo or were shown to segregate with disease

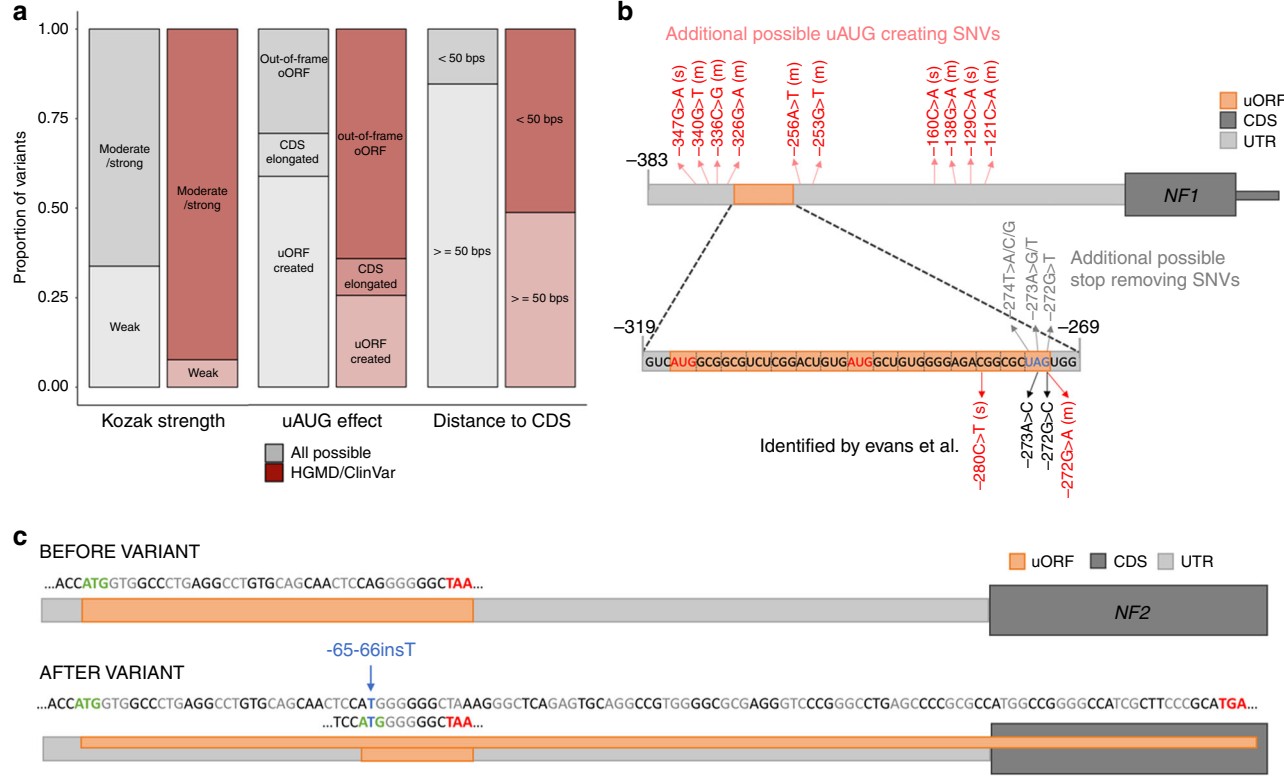

**Fig. 3** The role of uAUG-creating and uORF stop-removing variants in disease. **a** The proportion of 39 uAUG variants observed in HGMD and ClinVar (red bars) that fit into different sub-categories compared to all possible uAUG-creating SNVs (grey bars) in the same genes ($n = 1022$). Compared to all possible uAUG-creating variants, uAUG-creating variants observed in HGMD/ClinVar were significantly more likely to be created into a moderate or strong Kozak consensus (binomial $P = 3.5 \times 10^{-4}$), create an out-of-frame oORF (binomial $P = 1.1 \times 10^{-5}$), and be within 50 bp of the CDS (binomial $P = 3.9 \times 10^{-7}$). **b** Schematic of the *NF1* 5'UTR (light grey) showing the location of an existing uORF (orange) and the location of variants previously identified in patients with neurofibromatosis[29] in dark red (uAUG-creating) and black (stop-removing). uAUG-creating variants are annotated with the strength of the surrounding Kozak consensus in brackets ("s" for strong and "m" for moderate). All four published variants result in formation of an oORF out-of-frame with the CDS. Also annotated are the positions of all other possible uAUG-creating variants (light red; strong and moderate Kozak only), and stop-removing variants (grey) that would also create an out-of-frame oORF. **c** Schematic of the *NF2* 5'UTR (grey) showing the effects of the −65-66insT variant. The reference 5'UTR contains a uORF with a strong Kozak start site. Although the single-base insertion creates a novel uAUG which could be a new uORF start site, it also changes the frame of the existing uORF, so that it overlaps the CDS out-of-frame (forms an oORF). We predict this is the most likely mechanism of pathogenicity. CDS coding sequence, uORF upstream open reading frame, oORF overlapping open reading frame, HGMD the human gene mutation database

in the family (Supplementary Fig. 3a). While uAUG creation was proposed as the mechanism behind two of these variants, we now show that the other two variants both disrupt the stop codon of an existing uORF, resulting in an oORF which is out-of-frame with the CDS. This existing uORF has two start sites, both with strong Kozak consensus, and has prior evidence of active translation[24]. In Fig. 3b, we show these four variants along with an additional six stop-removing and ten uAUG-creating variants that would be predicted to also cause neurofibromatosis type 1 through the same mechanism if observed. In addition to these sixteen SNVs, indels that create high-impact uAUGs (oORF creating with strong/moderate Kozak consensus) or that cause a frameshift within the sequence of the existing uORF, resulting in an oORF, would also be predicted to cause disease.

A second example is *IRF6*, where three uAUG-creating variants have been identified in seven patients with Van de Woude syndrome[30,31]. These variants all arise in the context of a strong or moderate Kozak consensus and result in an out-of-frame oORF. There are nine additional possible uAUG-creating variants that would be predicted to yield the same effect in *IRF6* (Supplementary Fig. 4), suggesting it would be prudent to screen Van de Woude patients across all twelve sites.

**Genes where perturbing uORFs is likely important in disease**. To guide the research and clinical identification of uAUG-creating and stop-removing variants (referred to collectively as uORF-perturbing variants), we set about identifying genes where these variants are likely to be of high importance. Investigating 17,715 genes with annotated 5′UTRs and at least one possible uORF-perturbing variant, we first identified 4,986 genes where uORF-perturbing variants are unlikely to be deleterious: genes with existing oORFs (strong/moderate Kozak or evidence of translation), with predicted high-impact uORF-perturbing SNVs of appreciable frequency in gnomAD (>0.1%), with no possible high-impact uORF-perturbing SNVs, or that are tolerant to LoF (see methods; Supplementary Fig. 5a). Interestingly, these genes include 453 LoF intolerant (14.2% of most constrained LOEUF sextile) and 163 curated haploinsufficient or LoF disease genes (14.6%). Of the remaining 12,729 genes considered, 3191 (25.1%) are LoF-intolerant, known haploinsufficient or LoF disease genes and hence are genes where uORF-perturbing variants have a high likelihood of being deleterious (Fig. 4a). Despite only 18.0% of all classified genes falling into this high likelihood category (19.0% of all UTR bases when accounting for UTR length), 79% of uORF-perturbing variants in HGMD and ClinVar are found upstream of these genes (Fisher's $P = 1.6 \times 10^{-9}$; Fig. 4b).

There are 296 genes that have at least 10 possible high-impact uORF-perturbing SNVs, and for which LoF and/or haploinsufficiency is a known mechanism of human disease (either curated as haploinsufficient, curated as acting via a LoF mechanism in DDG2P or with ≥ 10 high-confidence pathogenic LoF variants documented in ClinVar), including both *IRF6* and *NF1*. We predict these to be a fruitful set to search for additional disease-causing uORF-perturbing variants (Supplementary Data 1; Supplementary Fig. 5b). To aid in the identification of uORF-perturbing variants we have created plugin for the Ensembl Variant Effect Predictor (VEP)[32] which annotates variants for predicted effects on translational regulation (available at https://github.com/ImperialCardioGenetics/uORFs).

**A novel uORF frameshift causes neurofibromatosisN type 2**. We analysed targeted sequencing data from a cohort of 1134 unrelated individuals diagnosed with neurofibromatosis type 2, which is caused by LoF variants in one of these prioritised genes, *NF2*. We identified a single 5′UTR variant in two unrelated

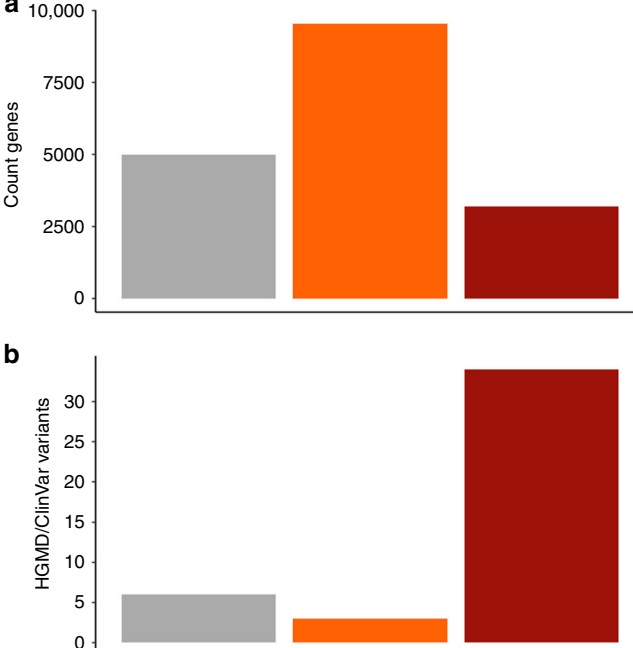

**Fig. 4** Identifying genes where uORF creating or disrupting variants are likely to have a role in disease. Genes were split into three distinct categories representing a 'low', 'moderate' and 'high' likelihood that uORF-perturbing variants are important. Low likelihood genes include those with existing oORFs, common (>0.1%) oORF creating variants in gnomAD or that are tolerant to LoF. Those in the high likelihood category are remaining genes that are LoF-intolerant or where haploinsufficient or LoF is a known disease mechanism (see methods). **a** The number of genes in each of the three categories. **b** The number of uAUG-creating and uORF stop-removing variants in HGMD upstream of genes in each category. Although only 19.2% of all classified genes fall into the high likelihood category (21.4% of all UTR bases when adjusting for UTR length), 83.7% of uORF-perturbing variants identified in HGMD and ClinVar are found upstream of these genes (Fisher's $P = 1.4 \times 10^{-19}$)

probands in this cohort (ENST00000338641:−66-65insT; GRCh37: chr22:29999922A >AT) that segregates with disease in three additional affected relatives across the two families (Supplementary Fig. 3b; Supplementary Table 3). This variant could act through two distinct uORF-disrupting mechanisms. While the insertion does create a new uAUG (in the context of a moderate Kozak consensus) an in-frame stop codon after only three codons would suggest only a weak effect on CDS translation. However, the *NF2* UTR contains an existing uORF with prior evidence of translation[24] and a strong Kozak consensus. The observed insertion changes the frame of this existing uORF, causing it to bypass the downstream stop codon and create an out-of-frame oORF (Fig. 3c). This oORF is predicted to lower translation of *NF2*, consistent with the known LoF disease mechanism, however, functional follow-up is required to confirm this hypothesis.

**Discussion**

We used data from 15,708 whole human genomes to explore the global impact of variants that create or perturb uORFs in 5′UTRs, which can lead to altered translation of the downstream protein. We show that creating a new uORF and hence initiating translation from an uAUG is an important regulatory mechanism. Our data suggest that the major underlying mechanism of

translational repression by uORFs is likely to be through competitive translation, since it is unlikely that novel peptides produced by uAUG-creating variants will be functional, and the most deleterious types of uAUG-creating and stop-removing variants are those that form oORFs.

Selective pressure on strongly translated uORFs has maintained features that promote re-initiation and prevent constitutive translational repression. Specifically, existing uORFs are selected to be short, further from the CDS, and to lack strong Kozak sequences[2]. This is in agreement with our results, which show a strongly skewed frequency spectrum for observed variants predicted to strongly inhibit translation, and an over-representation of these deleterious variants in disease cases.

We have defined a new category of variants, high-impact uORF-perturbing variants, a subset of which are likely to act as LoF by severely impacting translation. This class contains 145,398 possible SNVs (110,357 uAUG-creating and 35,041 stop-removing) across the genome, which are predicted to form oORFs from an uAUG with a strong or moderate Kozak consensus, or with prior evidence of translation. Of these, 3213 (2.2%) are observed in the whole genome sequence data from gnomAD. In addition, uAUG-creating insertions and deletions or frameshifts that transform existing uORFs into oORFs would also be predicted to have a high impact.

Whilst uORF-perturbing variants resulting in constitutive translational repression are likely to have LoF effects, the complex mechanisms of translational regulation including leaky scanning, re-initiation and the existence of internal ribosome entry sites makes it difficult to confidently predict the functional consequences of individual variants. Even variants predicted to be of high-impact may only result in partial LoF, reducing power to identify significant signals of selection. Confident interpretation of variants for a role in disease will require functional studies to assess the downstream impact of these variants on protein levels and/or additional genetic evidence, such as de novo occurrence or segregation with disease. It will also be interesting to study the impact of uORF-perturbing variants causing partial LoF on coding variant penetrance and their role in common disease phenotypes.

Even at a sample size of 15,708 individuals, we had limited power to observe uORF-perturbing variants, given their very small genomic footprint. Despite this, we identified specific genes such as *NF1*, *NF2* and *IRF6*, where uORF perturbation appears to be an important disease mechanism. In anticipation of future studies with much larger cohorts of WGS cases, we have identified a set of genes where there is a high likelihood that this mechanism will contribute to disease. This will also be useful for rare disease diagnosis, where even if WGS is undertaken this class of pathogenic variation is likely not evaluated and under-diagnosed.

In this work, we used variant frequencies in a large population dataset to study the global impact of a specific class of non-coding variants with a predicted functional effect. Previous studies using non-coding constraint have focused on entire regulatory regions[33] or concentrated exclusively on splicing[34,35]. These and other studies[36] have concluded that signals of constraint and selection are likely confined to individual bases[33] and diluted out when studying larger regions. Our results support this assertion; as the signal of negative selection associated with all UTR variants is not discernible from synonymous variants. We show the power of grouping individual non-coding bases by functional effect to identify subsets of variants with strong signals of selection.

## Methods

**Ethics statement**. We have complied with all relevant ethical regulations. This study was overseen by the Broad Institute's Office of Research Subject Protection and the Partners Human Research Committee, and was given a determination of Not Human Subjects Research. Informed consent was obtained from all participants.

**Study dataset**. We used the 15,708 whole genome sequenced individuals from version 2.1.1 of the Genome Aggregation Database (gnomAD), which is fully described in our companion paper[22]. These data were downloaded from https://gnomad.broadinstitute.org/downloads and queried using Hail version 0.2 (https://hail.is).

**Definition of 5'UTRs**. The start and end positions and sequence of the 5'UTRs of all protein-coding genes were downloaded from Ensembl biomart (Human genes GRCh37.p13) and filtered to only include canonical transcripts. Genes with no annotated 5'UTR on the canonical transcript were removed.

**Identification and classification of uAUG-creating variants**. Reading through each UTR from start to end (5' to 3'), we identified all instances where a SNV would create an ATG. We recorded the positions of all possible stop codons (TAA, TGA and TAG) and annotated each uAUG-creating variant with whether or not there was an in-frame stop codon within the UTR. To annotate the strength of the Kozak consensus into which the uAUG was formed we assessed the positions at −3 and +3 relative to the A of the AUG, known to be the most important bases for dictating strength of translation. If both the −3 base was either A or G and the +3 was G, Kozak was annotated as 'Strong', if either of these conditions was true, Kozak was deemed to be 'Moderate' and if neither was the case 'Weak'. uAUG-creating variants were also annotated with the distance to, and the frame relative to the coding sequence (CDS).

**Identification and classification of stop-removing variants**. Existing uORFs were defined as the combination of an ATG and in-frame stop codon (TAA, TGA or TAG) within a UTR. Each predicted uORF was annotated with the positions of all alternative downstream in-frame stop codons within the UTR and with the frame relative to the coding sequence. The Kozak strength of each uORF was defined as outlined above for uAUG-creating variants. Where multiple uAUGs converge on the same stop codon, the uORF is annotated with the strongest Kozak consensus. To identify uORFs with prior evidence of translation we downloaded all human small open reading frames (sORFs) from sorfs.org, a public repository of sORFs identified in humans, mice and fruit flies using ribosome profiling[24]. Predicted uORFs were marked as having prior evidence if the annotated stop codon matched an entry from sorfs.org.

Stop-removing variants were identified as SNVs that would change the base of a stop codon to any sequence that would not retain the stop (i.e., did not create another of TAA, TGA or TAG).

**Calculating MAPS**. For each set of variants we computed the mutability adjusted proportion of singletons, or MAPS. The basis of this approach has previously been described[23]. Briefly, for each substitution, accounting for 1 base of surrounding context (e.g., ACG ->ATG), we calculated the proportion of all possible variants (−3.9885 < GERP < 2.6607, 15 × < gnomAD coverage < 60 ×) that are observed in intergenic/intronic autosomal regions in a downsampled set of 1000 gnomAD whole-genomes. For C > T changes at CpG sites, variant proportions are calculated separately for three distinct bins of methylation. These proportions are then scaled so that the weighted genome-wide average is the human per-base, per-generation mutation rate (1.2e−8). The creation of these context-dependent mutation rates is described in more detail in our companion paper[22].

To determine the transformation between these mutation rates and the expected proportion of singletons, for each substitution and context (and methylation bin for CpGs), we regress the mutation rates against the observed proportion of singletons for synonymous variants. We use synonymous as a relatively neutral class of variants which should not be subject to any biases being investigating in UTRs, but that are distinct from bases used to define the model.

For a given list of possible variants, annotated with gnomAD allele counts using Hail (https://hail.is), we take only those that are observed in gnomAD and annotate each with the transformed mutation rate given the variant context (which now corresponds to the expected chance this site will be a singleton), and sum these values across the entire variant list to give an expected number of singletons. Variants are excluded if they are outliers on coverage in gnomAD (15× <coverage <60×), were found on the X or Y chromosome, or were filtered out of the gnomAD whole genomes.

Finally, this expected number of singletons is compared to the number of sites that are observed as singletons in gnomAD, to estimate MAPS.

$$\text{MAPS} = (\text{observed singletons} - \text{expected singletons})/\text{total observed variants}$$

(1)

Confidence intervals were calculated using bootstrapping. For a list of *n* observed variants, *n* variant sites are sampled at random with replacement and used to calculate MAPS. This is repeated over 10,000 permutations before the 5th

and 95th percentiles of the resulting MAPS distribution are taken as confidence intervals.

*P*-values we calculated using the same bootstrapping approach but for each permutation MAPS was calculated for each of the two variant sets of interest, A and B. The *P*-value was defined as the proportion of permutations where MAPS of B was less than MAPS of A.

$$P = \Sigma[(MAPS(B) - MAPS(A)) < 0]/permutations \qquad (2)$$

For coding variants, MAPS was calculated using the predicted impact on the canonical transcript.

**Using PhyloP to assess base-level conservation**. Per-base vertebrate PhyloP scores were extracted from the Combined Annotation Dependent Depletion (CADD) version v1.4 GRCh37 release files and used to annotate lists of all possible coding, UTR and uORF stop bases. To remove biases due to gene context and distance from the coding sequence, we created a set of matched UTR bases which comprised the 3 bases immediately upstream and downstream of the stop. Conserved bases were defined as those with PhyloP > = 2. We also checked for a significant difference between the entire distribution of scores using a Wilcoxon rank sum test for all stop-removing compared to matched UTR bases ($P = 8.1 \times 10^{-9}$).

**Identifying disease gene lists**. Developmental disease genes were downloaded from The Developmental Disorders Genotype-Phenotype Database (DDG2P) on the 6th October 2018. We included only genes categorised as 'confirmed' or 'probable'. Genes with a known dominant LoF mechanism were identified using the 'allelic requirement' and 'mutation consequence' annotations.

Genes intolerant and tolerant to LoF variants were identified using data from Karczewski et al. 2019[22]. Genes were ordered by their loss-of-function observed/ expected upper bound fraction (LOEUF) scores and the top and bottom sextiles were categorised as tolerant and intolerant, respectively.

We downloaded data from The Clinical Genome Resource (ClinGen) Dosage Sensitivity Map on 21st January 2019 (https://www.ncbi.nlm.nih.gov/projects/ dbvar/clingen/). Genes manually curated as haploinsufficient were defined as those with a score of 3 (sufficient evidence). In addition, we added genes curated as severe or moderately haploinsufficient by the MacArthur lab (https://github.com/ macarthur-lab/gene_lists/tree/master/lists).

**Searching for uORF-perturbing variants in HGMD and ClinVar**. Lists of all possible uAUG-creating and stop-removing SNVs were intersected with all DM variants from HGMD pro release 2018.1 and all ClinVar Pathogenic or Likely Pathogenic variants from the August 2018 release (clinvar_20180805.vcf). In addition, we created a list of all possible 1–5 bp deletions that would create an uAUG, annotated as described for SNVs above, and also searched for these variants. We did not investigate small insertions or deletion >5 bps due to the inhibitory number of possible variants.

**Sub-classifying genes**. uAUG-creating variants were classified as 'high-impact' if they are formed into a high or moderate Kozak consensus and if they either form an oORF or result in transcript elongation. Stop removing variants were similarly classified as 'high-impact' if the original uORF start site has a strong or moderate Kozak and/or the uORF is documented in sorfs.org and the variants results in a oORF or a transcript elongation.

Genes were divided into nine categories according to the following logic.

Class 0–genes with no annotated 5'UTR on the canonical transcript.

Class 1–genes with no possible uAUG-creating or stop-removing SNVs identified.

Class 2–remaining genes with no possible SNVs of predicted high-impact.

Class 3–remaining genes where the UTR has a high-confidence oORF (strong/ moderate Kozak or documented in sorfs.orf) indicating creating a second would be of low-impact.

Class 4–remaining genes where one or more identified high-impact SNVs have AF > 0.1% in gnomAD (genomes AC >15).

Class 5–remaining genes that are tolerant to LoF variants.

Class 8–remaining genes curated as haploinsufficient by ClinGen or the MacArthur lab, curated as acting via a loss-of-function mechanism in DDG2P or with > = 10 high-confidence Pathogenic LoF variants in ClinVar (known LoF disease genes).

Class 7–remaining genes intolerant to LoF variants or with > = 2 high-confidence Pathogenic LoF variants in ClinVar.

Class 6–all genes not classified in any other class.

The nine gene classes were grouped into three categories corresponding to low (classes 2, 3, 4 and 5), moderate (class 6) and high (classes 7 and 8) likelihood that high-impact uORF-perturbing variants would have a deleterious effect.

**Sequencing of individuals with neurofibromatosis type 2**. A cohort of 1134 unrelated individuals with neurofibromatosis type 2 were recruited to the Centre for Genomic Medicine at St Mary's Hospital, Manchester. All individuals fulfilled both Manchester and NIH criteria for diagnosis. Ethical approval for use of these samples and anonymised associated clinical data was obtained from the North West Greater Manchester Central Research Ethics Committee (reference 10/ H1008/74). Informed consent was obtained from all participants. All patients were sequenced across the *NF2* gene. Two individuals were identified to carry a single 5'UTR variant (ENST00000338641:-66-65insT; GRCh37:chr22:29999922A >AT). Both carriers were found to have no variants in *SMARCB1* or *LZTRA1* and no coding variants in *NF2*. The -66-65insT variant segregated with disease in 3 affected siblings in one family and in affected parent and child in another (Supplementary Fig. 3b). Phenotypic data for all family members can be found in Supplementary Table 3.

**Reporting summary**. Further information on research design is available in the Nature Research Reporting Summary linked to this article.

## Data availability
All possible uAUG-creating and stop-removing SNVs for canonical Gencode transcripts along with likelihood classifications for all genes are available for download at https:// github.com/ImperialCardioGenetics/uORFs.

## Code availability
To aid in the identification of uORF-perturbing variants we have created a VEP plugin which annotates variants for predicted effects on translational regulation. This script, along with those used to analyse the data and create figures for the manuscript, is freely available at https://github.com/ImperialCardioGenetics/uORFs.

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

## Acknowledgements

N.W. is supported by a Rosetrees and Stoneygate Imperial College Research Fellowship. This work was supported by the Wellcome Trust (107469/Z/15/Z), the Medical Research Council (UK), the NIHR Biomedical Research Unit in Cardiovascular Disease at Royal Brompton and Harefield NHS Foundation Trust and Imperial College London, the NIHR Imperial College Biomedical Research Centre, the Fondation Leducq (11 CVD-01), a Health Innovation Challenge Fund award from the Wellcome Trust and Department of Health, UK (HICF-R6–373), and by NIDDK U54DK105566 and NIGMS R01GM104371. D.G.E. and M.J.S. are funded by the NIHR Biomedical research centre Manchester (IS-BRC-1215-20007). L.C.F. is supported by the Swiss National Science Foundation (Advanced Postdoc.Mobility 177853). N.Q. is supported by the Imperial College Academic Health Science Centre. The results published here are in part based upon data: (1) generated by The Cancer Genome Atlas managed by the NCI and NHGRI (accession: phs000178.v10.p8). Information about TCGA can be found at http://cancergenome.nih.gov, (2) generated by the Genotype-Tissue Expression Project (GTEx) managed by the NIH Common Fund and NHGRI (accession: phs000424.v7.p2), (3) generated by the Exome Sequencing Project, managed by NHLBI, (4) generated by the Alzheimer's Disease Sequencing Project (ADSP), managed by the NIA and NHGRI (accession: phs000572.v7.p4). The views expressed in this work are those of the authors and not necessarily those of any of the funders. We would like to thank Dr. Christopher M Yates for his statistical advice.

## Author contributions

N.W., D.G.M. and J.S.W designed the study and supervised the work. N.W., K.J.K., X.Z. and S.C. conducted analyses. M.J.S, D.G.E, gnomAD PG and gnomAD C provided data for the study. A.M.R., N.M.Q., S.S., O.R., J.A., A.H.O.-L., L.C.F., S.A.C. and P.J.R.B. provided critical feedback on all analyses and figures. The paper was written by N.W., D.G.M. and J.S.W. with comments provided by all other authors.

## Additional information

**Competing interests:** D.G.M. is a founder with equity in Goldfinch Bio. The remaining authors declare no competing interests.

## Genome Aggregation Database Production Team

Irina M. Armean[3,4,12], Eric Banks[13], Louis Bergelson[13], Kristian Cibulskis[13], Ryan L. Collins[3,14,15], Kristen M. Connolly[16], Miguel Covarrubias[13], Beryl Cummings[3,4,17], Mark J. Daly[3,4,18], Stacey Donnelly[3], Yossi Farjoun[13], Steven Ferriera[19], Stacey Gabriel[19], Laura D. Gauthier[13], Jeff Gentry[13], Namrata Gupta[3,19], Thibault Jeandet[13], Diane Kaplan[13], Kristen M. Laricchia[3,4], Christopher Llanwarne[13], Eric V. Minikel[3], Ruchi Munshi[13], Benjamin M. Neale[3,4], Sam Novod[13], Nikelle Petrillo[13], Timothy Poterba[3,4,18], David Roazen[13], Valentin Ruano-Rubio[13], Andrea Saltzman[3], Kaitlin E. Samocha[20], Molly Schleicher[3], Cotton Seed[4,18],

Matthew Solomonson[3,4], Jose Soto[13], Grace Tiao[3,4], Kathleen Tibbetts[13], Charlotte Tolonen[13], Christopher Vittal[4,18], Gordon Wade[13], Arcturus Wang[3,4,18], Qingbo Wang[3,4,15], Nicholas A. Watts[3,4] & Ben Weisburd[13]

[12]European Molecular Biology Laboratory, European Bioinformatics Institute, Wellcome Genome Campus, Hinxton, Cambridge CB10 1SD, UK. [13]Data Sciences Platform, Broad Institute of MIT and Harvard, Cambridge, MA 02142, USA. [14]Center for Genomic Medicine, Massachusetts General Hospital, Boston, MA 02114, USA. [15]Program in Bioinformatics and Integrative Genomics, Harvard Medical School, Boston, MA 02115, USA. [16]Genomics Platform, Broad Institute of MIT and Harvard, Cambridge, MA 02142, USA. [17]Program in Biological and Biomedical Sciences, Harvard Medical School, Boston, MA 02115, USA. [18]Stanley Center for Psychiatric Research, Broad Institute of MIT and Harvard, Cambridge, MA 02142, USA. [19]Broad Genomics, Broad Institute of MIT and Harvard, Cambridge, MA 02142, USA. [20]Wellcome Sanger Institute, Wellcome Genome Campus, Hinxton, Cambridge CB10 1SA, UK

## Genome Aggregation Database Consortium

Carlos A. Aguilar Salinas[21], Tariq Ahmad[22], Christine M. Albert[23,24], Diego Ardissino[25], Gil Atzmon[26,27], John Barnard[28], Laurent Beaugerie[29], Emelia J. Benjamin[30,31,32], Michael Boehnke[33], Lori L. Bonnycastle[34], Erwin P. Bottinger[35], Donald W. Bowden[36,37,38], Matthew J. Bown[39,40], John C. Chambers[41,42,43], Juliana C. Chan[44], Daniel Chasman[23,45], Judy Cho[35], Mina K. Chung[46], Bruce Cohen[45,47], Adolfo Correa[48], Dana Dabelea[49], Mark J. Daly[3,4,18], Dawood Darbar[50], Ravindranath Duggirala[51], Josée Dupuis[52,53], Patrick T. Ellinor[3,54], Roberto Elosua[55,56,57], Jeanette Erdmann[58,59,60], Tõnu Esko[3,61], Martti Färkkilä[62], Jose Florez[63], Andre Franke[64], Gad Getz[45,65,66], Benjamin Glaser[67], Stephen J. Glatt[68], David Goldstein[69,70], Clicerio Gonzalez[71], Leif Groop[72,73], Christopher Haiman[74], Craig Hanis[75], Matthew Harms[76,77], Mikko Hiltunen[78], Matti M. Holi[79], Christina M. Hultman[80,81], Mikko Kallela[82], Jaakko Kaprio[73,83], Sekar Kathiresan[45,84,85], Bong-Jo Kim[86], Young Jin Kim[86], George Kirov[87], Jaspal Kooner[42,43,88], Seppo Koskinen[89], Harlan M. Krumholz[90], Subra Kugathasan[91], Soo Heon Kwak[92], Markku Laakso[93,94], Terho Lehtimäki[95], Ruth J.F. Loos[35,96], Steven A. Lubitz[3,56], Ronald C.W. Ma[44,97,98], Jaume Marrugat[58,99], Kari M. Mattila[95], Steven McCarroll[51,100], Mark I. McCarthy[101,102,103], Dermot McGovern[104], Ruth McPherson[105], James B. Meigs[45,106,107], Olle Melander[108], Andres Metspalu[61], Benjamin M. Neale[3,4], Peter M. Nilsson[109], Michael C. O'Donovan[87], Dost Ongur[45,47], Lorena Orozco[110], Michael J. Owen[87], Colin N.A. Palmer[111], Aarno Palotie[4,51,73], Kyong Soo Park[92,112], Carlos Pato[113], Ann E. Pulver[114], Nazneen Rahman[115], Anne M. Remes[116], John D. Rioux[117,118], Samuli Ripatti[73,83,119], Dan M. Roden[120,121], Danish Saleheen[122,123,124], Veikko Salomaa[125], Nilesh J. Samani[39,40], Jeremiah Scharf[3,18,84], Heribert Schunkert[126,127], Moore B. Shoemaker[128], Pamela Sklar[129,130,131], Hilkka Soininen[132], Harry Sokol[29], Tim Spector[133], Patrick F. Sullivan[80,134], Jaana Suvisaari[125], E. Shyong Tai[135,136,137], Yik Ying Teo[136,138,139], Tuomi Tiinamaija[73,140,141], Ming Tsuang[142,143], Dan Turner[144], Teresa Tusie-Luna[145,146], Erkki Vartiainen[83], Hugh Watkins[147], Rinse K. Weersma[148], Maija Wessman[73,140], James G. Wilson[149] & Ramnik J. Xavier[150,151]

[21]Unidad de Investigacion de Enfermedades Metabolicas, Instituto Nacional de Ciencias Medicas y Nutricion, Mexico City, Mexico. [22]Peninsula College of Medicine and Dentistry, Exeter EX2 4TH, UK. [23]Division of Preventive Medicine, Brigham and Women's Hospital, Boston, MA 02115, USA. [24]Division of Cardiovascular Medicine, Brigham and Women's Hospital and Harvard Medical School, Boston, MA 02115, USA. [25]Department of Cardiology University Hospital, 43100 Parma, Italy. [26]Department of Biology, Faculty of Natural Sciences, University of Haifa, 3498838 Haifa, Israel. [27]Departments of Medicine and Genetics, Albert Einstein College of Medicine, Bronx, NY 10461, USA. [28]Department of Quantitative Health Sciences, Lerner Research Institute, Cleveland Clinic, Cleveland, OH 44122, USA. [29]APHP, Gastroenterology Department, Saint Antoine Hospital, Sorbonne Université, Paris, France. [30]NHLBI and Boston University's Framingham Heart Study, Framingham, MA 01701, USA. [31]Department of Medicine, Boston University School of Medicine, Boston, MA 02118, USA. [32]Department of Epidemiology, Boston University School of Public Health, Boston, MA 02118, USA. [33]Department of Biostatistics and Center for Statistical Genetics, University of Michigan, Ann Arbor, MI 48109, USA. [34]National Human Genome Research Institute, National Institutes of Health, Bethesda, MD 20892, USA. [35]The Charles Bronfman Institute for Personalized Medicine, Icahn School of Medicine at Mount Sinai, New York, NY 10029, USA. [36]Department of Biochemistry, Wake Forest School of Medicine, Winston-Salem, NC 27101, USA. [37]Center for Genomics and Personalized Medicine Research, Wake Forest School of Medicine, Winston-Salem, NC 27101, USA. [38]Center for Diabetes Research, Wake Forest School of Medicine, Winston-Salem, NC 27101, USA. [39]Department of Cardiovascular Sciences, University of Leicester, Leicester LE1 7RH, UK. [40]NIHR Leicester Biomedical Research Centre, Glenfield Hospital, Leicester LE3 9QP, UK. [41]Department of Epidemiology and Biostatistics, Imperial College London, London SW7 2BX, UK. [42]Department of

Cardiology, Ealing Hospital NHS Trust, Southall UB1 3HW, UK. [43]Imperial College Healthcare NHS Trust, Imperial College London, London SW7 2BX, UK. [44]Department of Medicine and Therapeutics, The Chinese University of Hong Kong, 999077 Hong Kong, China. [45]Department of Medicine, Harvard Medical School, Boston, MA 02115, USA. [46]Departments of Cardiovascular Medicine, Cellular and Molecular Medicine, Molecular Cardiology, and Quantitative Health Sciences, Cleveland Clinic, Cleveland, OH 44195, USA. [47]McLean Hospital, Belmont, MA 02478, USA. [48]Department of Medicine, University of Mississippi Medical Center, Jackson, MS 39216, USA. [49]Department of Epidemiology, Colorado School of Public Health, Aurora, Colorado 80246, USA. [50]Department of Medicine and Pharmacology, University of Illinois at Chicago, Chicago, IL 60607, USA. [51]Department of Genetics, Texas Biomedical Research Institute, San Antonio, TX 78227, USA. [52]Department of Biostatistics, Boston University School of Public Health, Boston, MA 02118, USA. [53]National Heart, Lung, and Blood Institute's Framingham Heart Study, Framingham, MA 01702, USA. [54]Cardiac Arrhythmia Service and Cardiovascular Research Center, Massachusetts General Hospital, Boston, MA 02114, USA. [55]Cardiovascular Epidemiology and Genetics, Hospital del Mar Medical Research Institute (IMIM), Barcelona, Catalonia, Spain. [56]CIBER CV, Barcelona, Catalonia, Spain. [57]Departament of Medicine, Medical School, University of Vic-Central University of Catalonia, Catalonia 08500, Spain. [58]Institute for Cardiogenetics, University of Lübeck, Lübeck 23562, Germany. [59]DZHK (German Research Centre for Cardiovascular Research) partner site Hamburg/Lübeck/Kiel, 23562 Lübeck, Germany. [60]University Heart Center Lübeck, 23562 Lübeck, Germany. [61]Estonian Genome Center, Institute of Genomics, University of Tartu, Tartu 50090, Estonia. [62]Clinic of Gastroenterology, Helsinki University and Helsinki University Hospital, FL 00014 Helsinki, Finland. [63]Diabetes Unit and Center for Genomic Medicine, Massachusetts General Hospital; Programs in Metabolism and Medical and Population Genetics, Broad Institute; Department of Medicine, Harvard Medical School, Boston, MA 02115, USA. [64]Institute of Clinical Molecular Biology (IKMB), Christian-Albrechts-University of Kiel, Kiel 24118, Germany. [65]Bioinformatics Program, Department of Pathology, MGH Cancer Center, Boston 02114, USA. [66]Cancer Genome Computational Analysis, Broad Institute, Cambridge 02142, USA. [67]Endocrinology and Metabolism Department, Hadassah-Hebrew University Medical Center, Jerusalem, Israel. [68]Department of Psychiatry and Behavioral Sciences, SUNY Upstate Medical University, Syracuse, NY 13421, USA. [69]Institute for Genomic Medicine, Columbia University Medical Center, Hammer Health Sciences, 1408, 701 West 168th Street, New York, NY 10032, USA. [70]Department of Genetics & Development, Columbia University Medical Center, Hammer Health Sciences, 1602, 701 West 168th Street, New York, NY 10032, USA. [71]Centro de Investigacion en Salud Poblacional Instituto Nacional de Salud Publica MEXICO, Cuernavace 62100, Mexico. [72]Lund University, SE-221 00 Lund, Sweden. [73]Institute for Molecular Medicine Finland (FIMM), HiLIFE, University of Helsinki, FL 00014 Helsinki, Finland. [74]Lund University Diabetes Centre, SE-221 00 Lund, Sweden. [75]Human Genetics Center, University of Texas Health Science Center at Houston, Houston, TX 77030, USA. [76]Department of Neurology, Columbia University, New York 10032, USA. [77]Institute of Genomic Medicine, Columbia University, New York 10032, USA. [78]Institute of Biomedicine, University of Eastern Finland, Kuopio FI-80101, Finland. [79]Department of Psychiatry, PL 320, Helsinki University Central Hospital, Lapinlahdentie, 00 180, Helsinki, Finland. [80]Department of Medical Epidemiology and Biostatistics, Karolinska Institutet, Stockholm 171 77, Sweden. [81]Icahn School of Medicine at Mount Sinai, New York, NY 10029, USA. [82]Department of Neurology, Helsinki University Central Hospital, Fl 00290 Helsinki, Finland. [83]Department of Public Health, Faculty of Medicine, University of Helsinki, FL 00014 Helsinki, Finland. [84]Center for Genomic Medicine, Massachusetts General Hospital, Boston, MA 02114, USA. [85]Cardiovascular Disease Initiative and Program in Medical and Population Genetics, Broad Institute of MIT and Harvard, Cambridge, MA 02142, USA. [86]Center for Genome Science, Korea National Institute of Health, Chungcheongbuk-do, Republic of Korea. [87]MRC Centre for Neuropsychiatric Genetics and Genomics, Cardiff University School of Medicine, Hadyn Ellis Building, Maindy Road, Cardiff CF24 4HQ, UK. [88]National Heart and Lung Institute, Cardiovascular Sciences, Imperial College London, Hammersmith Campus, London SW7 2BX, UK. [89]Department of Health, THL-National Institute for Health and Welfare, 00271 Helsinki, Finland. [90]Section of Cardiovascular Medicine, Department of Internal Medicine, Yale School of Medicine, New Haven, CT 06510, USA. [91]Division of Pediatric Gastroenterology, Emory University School of Medicine, Atlanta, GA 30322, USA. [92]Department of Internal Medicine, Seoul National University Hospital, Seoul, Republic of Korea. [93]Institute of Clinical Medicine, The University of Eastern Finland, Kuopio, Finland. [94]Kuopio University Hospital, Kuopio 70210, Finland. [95]Department of Clinical Chemistry, Fimlab Laboratories and Finnish Cardiovascular Research Center-Tampere, Faculty of Medicine and Health Technology, Tampere University, FI-33014 Tampere, Finland. [96]The Mindich Child Health and Development Institute, Icahn School of Medicine at Mount Sinai, New York, NY 10029, USA. [97]Li Ka Shing Institute of Health Sciences, The Chinese University of Hong Kong, Hong Kong, China. [98]Hong Kong Institute of Diabetes and Obesity, The Chinese University of Hong Kong, Hong Kong, China. [99]Cardiovascular Research REGICOR Group, Hospital del Mar Medical Research Institute (IMIM), Barcelona 08003 Catalonia, Spain. [100]Department of Genetics, Harvard Medical School, Boston, MA 02115, USA. [101]Oxford Centre for Diabetes, Endocrinology and Metabolism, University of Oxford, Churchill Hospital, Old Road, Headington, Oxford OX3 7LJ, UK. [102]Wellcome Centre for Human Genetics, University of Oxford, Roosevelt Drive, Oxford OX3 7BN, UK. [103]Oxford NIHR Biomedical Research Centre, Oxford University Hospitals NHS Foundation Trust, John Radcliffe Hospital, Oxford OX3 9DU, UK. [104]F Widjaja Foundation Inflammatory Bowel and Immunobiology Research Institute, Cedars-Sinai Medical Center, Los Angeles, CA 90048, USA. [105]Atherogenomics Laboratory, University of Ottawa Heart Institute, Ottawa, ON K1Y 4W7, Canada. [106]Division of General Internal Medicine, Massachusetts General Hospital, Boston, MA 02114, USA. [107]Program in Population and Medical Genetics, Broad Institute, Cambridge, MA 02142, USA. [108]Department of Clinical Sciences, University Hospital Malmo Clinical Research Center, Lund University, SE-221 00 Malmo, Sweden. [109]Department of Clinical Sciences, Lund University, Skane University Hospital, SE-221 00 Malmo, Sweden. [110]Instituto Nacional de Medicina Genómica (INMEGEN), Mexico City 14610, Mexico. [111]Medical Research Institute, Ninewells Hospital and Medical School, University of Dundee, Dundee DD1 4HN, UK. [112]Department of Molecular Medicine and Biopharmaceutical Sciences, Graduate School of Convergence Science and Technology, Seoul National University, Seoul, Republic of Korea. [113]Department of Psychiatry, Keck School of Medicine at the University of Southern California, Los Angeles, CA 90033, USA. [114]Department of Psychiatry and Behavioral Sciences, Johns Hopkins University School of Medicine, Baltimore, MD 20105, USA. [115]Division of Genetics and Epidemiology, Institute of Cancer Research, London SM2 5NG, UK. [116]Medical Research Center, Oulu University Hospital, Oulu, Finland and Research Unit of Clinical Neuroscience, Neurology, University of Oulu, FI-90014 Oulu, Finland. [117]Research Center, Montreal Heart Institute, Montreal, QC H1T 1C8, Canada. [118]Department of Medicine, Faculty of Medicine, Université de Montréal, Montréal H3T 1J4 Québec, Canada. [119]Broad Institute of MIT and Harvard, Cambridge, MA 02142, USA. [120]Department of Biomedical Informatics, Vanderbilt University Medical Center, Nashville, TN 37212, USA. [121]Department of Medicine, Vanderbilt University Medical Center, Nashville, TN 37212, USA. [122]Department of Biostatistics and Epidemiology, Perelman School of Medicine at the University of Pennsylvania, Philadelphia, PA 19104, USA. [123]Department of Medicine, Perelman School of Medicine at the University of Pennsylvania, Philadelphia, PA 19104, USA. [124]Center for Non-Communicable Diseases, Karachi 75300, Pakistan. [125]National Institute for Health and Welfare, Helsinki FI 00271, Finland. [126]Deutsches Herzzentrum München, München 80636, Germany. [127]Technische Universität München, München 80333, Germany. [128]Division of Cardiovascular Medicine, Nashville VA Medical Center and Vanderbilt University, School of Medicine, Nashville, TN 37232-8802, USA. [129]Department of Psychiatry, Icahn School of Medicine at Mount Sinai, New York, NY 10029, USA. [130]Department of Genetics and Genomic Sciences, Icahn School of Medicine at Mount Sinai, New York, NY 10029, USA. [131]Institute for Genomics and Multiscale Biology, Icahn School of Medicine at Mount Sinai, New York, NY 10029, USA. [132]Institute of Clinical Medicine, neurology, University of Eastern Finland, Kuopio, Finland. [133]Department of Twin Research and Genetic Epidemiology, King's College London, London SE1 7EH, UK. [134]Departments

of Genetics and Psychiatry, University of North Carolina, Chapel Hill, NC 27599, USA. [135]Saw Swee Hock School of Public Health, National University of Singapore, National University Health System, Singapore 117549, Singapore. [136]Department of Medicine, Yong Loo Lin School of Medicine, National University of Singapore, Singapore 117597, Singapore. [137]Duke-NUS Graduate Medical School, Singapore 169857, Singapore. [138]Life Sciences Institute, National University of Singapore, Singapore 117456, Singapore. [139]Department of Statistics and Applied Probability, National University of Singapore, Singapore 119077, Singapore. [140]Folkhälsan Institute of Genetics, Folkhälsan Research Center, Helsinki, Finland. [141]HUCH Abdominal Center, Helsinki University Hospital, Helsinki 00260, Finland. [142]Center for Behavioral Genomics, Department of Psychiatry, University of California, San Diego, CA 94720, USA. [143]Institute of Genomic Medicine, University of California, San Diego, CA 94720, USA. [144]Juliet Keidan Institute of Pediatric Gastroenterology, Shaare Zedek Medical Center, The Hebrew University of Jerusalem, Jerusalem, Israel. [145]Instituto de Investigaciones Biomédicas UNAM, Mexico City 04510, Mexico. [146]Instituto Nacional de Ciencias Médicas y Nutrición Salvador Zubirán Mexico City, Mexico City 14080, Mexico. [147]Radcliffe Department of Medicine, University of Oxford, Oxford OX3 9DU, UK. [148]Department of Gastroenterology and Hepatology, University of Groningen and University Medical Center Groningen, Groningen 9713 GZ, The Netherlands. [149]Department of Physiology and Biophysics, University of Mississippi Medical Center, Jackson, MS 39216, USA. [150]Program in Infectious Disease and Microbiome, Broad Institute of MIT and Harvard, Cambridge, MA 02142, USA. [151]Center for Computational and Integrative Biology, Massachusetts General Hospital, Boston, MA 02114, USA

