## [Peer Review File · Nature Communications]

Reviewers' Comments:

Reviewer #1:

Remarks to the Author:

Whiffin et al. have presented an interesting analysis of the expansive gnomAD human whole genome sequence dataset that identifies thousands of variants that may be contributing to disease in a previously underappreciated way.

The novelty of this article lies in the use of the large dataset to actually test the deleteriousness of a specific class of non-coding SNPs previously speculated to be contributing to disease.

The authors define the uORF-perturbing classes of mutations, show that there is likely increased negative selection against these mutations, identify existing disease annotations as validation for the disease-causing potential of these mutations, and finally pinpoint hundreds of genes where these mutations are most likely to be disease-causing.

Specifically, they address a disease mechanism that does not get much attention "in either clinical or research settings, having been excluded from most exome capture target regions."

In defining this class of disease-causing variants, this work shows that large genomic datasets can be invaluable for the study of disease variation over simple GWAS style analyses where mechanisms are often afterthoughts. The authors realistically allocate the usefulness of their results: namely to identify candidates for further study. Additionally, they also provide a plugin to the commonly used Ensembl Variant Effect Predictor, to increase accessibility of their analyses to other researchers.

Overall, the manuscript is well written with clearly presented results and figures that are appropriate for the presentation of their results.

A minor comment follows:

Regarding the statement: "These results further illustrate the power of MAPS to identify variant classes most likely to be disease-causing." While this may be true, this statement feels out of place in the article, as the selling point of this work is not the MAPS method itself, but the disease-causing uORF-perturbers.

Reviewer #2:

Remarks to the Author:

This work by Whiffin and colleagues provides an excellent investigation into a long-standing ambiguity in the field about how best to interpret and interact with 5' UTR start gained annotations. The authors have carefully balanced both scientific and utility aspects; making their new predictions publicly accessible through working with VeIP team and on github. I have enjoyed reading this work and have only minor comments/suggestions to make:

* In the abstract and throughout the manuscript tallies are provided, but often as a reader I'd like to also understand what proportion those tallies reflect. e.g., 14,897 variants creating new start codons shown to be under strong negative selection - out of how many assessed 5' UTR start gain variants? Is that proportion significantly elevated compared to randomly sampling sites in the 5' UTR of the same class of genes?

* Earlier work (incl. PMID26332131) looking at the conservation of noncoding sequence across genes did find that dosage sensitive genes have a higher conserved UTR. To disentangle the current correlations with pLI and dosage sensitive genes, is there a way to potentially show whether 5' UTR start variant sites are found to be more highly conserved relative to the

corresponding gene's overall 5' UTR sequence?

* There is a growing catalogue of accessible pediatric disease trio WGS, including from the 100K genomes project and other large-scale developmental disorder collections. Often the de novo significance enrichment identified for these ID/DD genes is driven by LoF mechanisms. As such, is there an opportunity to construct a statistical framework to address whether there is an overall elevated rate in 5' UTR start variants among ID/DD trio cohorts given the total sequence context occupied by all possible such variants (562,196)? Something along these lines could considerably assist in understanding whether there is a significant contribution of this class of variants to overall ID/DD architecture. Perhaps focused on oORF, given its MAPS score relative to pLOF alleles.

* For the past year or so, TOPMED has represented a considerably larger resource of WGS than gnomAD. Were TOPMED 5' UTR variants considered?

Reviewer #3:

Remarks to the Author:

I was asked to review and comment on the pathogenicity claims pertaining to neurofibromatosis in this manuscript in light of the claimed 5' UTR variant deleteriousness. Although I read the entire manuscript and supplements, I focus especially on the claims related to neurofibromatosis. Overall, I found the work well-written, measured and plausible.

Neurofibromatosis type 1. There are four NF1 variants originally reported in Evans et al EBioMedicine 2016 and characterized in the current manuscript as uAUG-creating or stop-removing (Figure 3). I was a reviewer on the Evans 2016 paper, and thought it was a well-done study of NF1 variation. I re-examined the paper and the description of the four 5'-UTR NF1 variants (Table 4 of Evans 2016). There the authors note that non-specific NF1 features segregate with variation in two families; two other families had apparent de novo disease. No other report cataloged in HGMD describes such 5'-UTR variation. In NF1, claims of pathogenicity of 5'-UTR variation (or any non-splice regulatory region) are rare to non-existent. That said, it has been long-recognized that sequencing of NF1 detects >95% deleteriousness variation in people who meet NF1 diagnostic criteria, even when accounting for Legius syndrome. Certainly 5'-UTR variation may explain some of that "missing" <5% variation and this paper is a useful addition to identifying a plausible source of that variation.

Given the assertion that those four NF1 5'-UTR variants are pathogenic, the authors could strengthen their argument by documenting more clinical details from the families/individuals that harbored that variation, with pedigrees. (The original work was done in Manchester by Gareth Evans's group and I see that one of the authors is affiliated with St. Mary's in Manchester.) The authors should also verify and assert (if true) that no other NF1 variation (including deletion testing) was detected in those individuals. If possible, documentation of negative SPRED1 sequencing would be helpful, especially if the NF1 patients had pigmentary findings. If available, tumors with the putative 5'-UTR plus a somatic second hit would be additional convincing evidence.

Neurofibromatosis type 2. I have similar comments for NF2, however, the sequenced cohort has not been published. I would prefer a much more detailed accounting of the NF2 phenotype, with pedigrees. If available, tumors with the putative 5'-UTR plus a somatic second hit would be additional convincing evidence.

More minor comments:

- 1) Prefer either "neurofibromatosis type 1" instead of just "neurofibromatosis" when referring to the consequence of pathogenic variation in NF1.
- 2) In Supplemental Tables 1 and 2, please add HGMD accession numbers for given variants.

Although I was able to locate the referenced variation, it would have been much easier to do so if given the proper HGMD accession number. Even better would be the PubMed ID(s) for publications reporting particular variation. This way readers can quickly judge for themselves the phenotypic consequence of the uAUG-creating or stop-removing variants

We would like to thank the reviewers for their positive feedback and very useful comments on our manuscript. Below is our point-by-point response to their suggestions.

Reviewer #1 (Remarks to the Author):

Whiffin et al. have presented an interesting analysis of the expansive gnomAD human whole genome sequence dataset that identifies thousands of variants that may be contributing to disease in a previously underappreciated way.

The novelty of this article lies in the use of the large dataset to actually test the deleteriousness of a specific class of non-coding SNPs previously speculated to be contributing to disease.

The authors define the uORF-perturbing classes of mutations, show that there is likely increased negative selection against these mutations, identify existing disease annotations as validation for the disease-causing potential of these mutations, and finally pinpoint hundreds of genes where these mutations are most likely to be disease-causing.

Specifically, they address a disease mechanism that does not get much attention “in either clinical or research settings, having been excluded from most exome capture target regions.”

In defining this class of disease-causing variants, this work shows that large genomic datasets can be invaluable for the study of disease variation over simple GWAS style analyses where mechanisms are often afterthoughts. The authors realistically allocate the usefulness of their results: namely to identify candidates for further study. Additionally, they also provide a plugin to the commonly used Ensembl Variant Effect Predictor, to increase accessibility of their analyses to other researchers.

Overall, the manuscript is well written with clearly presented results and figures that are appropriate for the presentation of their results.

A minor comment follows:

Regarding the statement: “These results further illustrate the power of MAPS to identify variant classes most likely to be disease-causing.” While this may be true, this statement feels out of place in the article, as the selling point of this work is not the MAPS method itself, but the disease-causing uORF-perturbers.

Response: We have edited this sentence to read: “These results support the assertion that the variant classes identified by MAPS as under strongest negative selection are most likely to be disease causing.” We think this wording better reflects the results of this paragraph in terms of uORF-perturbing variants rather than focussing on the MAPS method.

Reviewer #2 (Remarks to the Author):

This work by Whiffin and colleagues provides an excellent investigation into a long-standing ambiguity in the field about how best to interpret and interact with 5' UTR start gained annotations. The authors have carefully balanced both scientific and utility aspects; making their new predictions publicly accessible through working with Ve!P team and on github. I have enjoyed reading this work and have only minor comments/suggestions to make:

** In the abstract and throughout the manuscript tallies are provided, but often as a reader I'd like to also understand what proportion those tallies reflect. e.g., 14,897 variants creating new start codons shown to be under strong negative selection - out of how many assessed 5' UTR start gain variants? Is that proportion significantly elevated compared to randomly sampling sites in the 5' UTR of the same class of genes?*

Response: Thank you very much for this comment. You have highlighted that this sentence in our abstract is unclear and doesn't reflect what we intended. We assessed all 14,797 autosomal high-quality start creating variants that are observed in gnomAD and found that this entire category of variants are under strong negative selection. We have now removed the numbers from the abstract to alleviate any confusion as to whether 14,897 represents a sub-set of variants. We have also made sure that the derivation of the 14,897 number is clear in the main text. Specifically the text reads "We identified all possible single nucleotide variants (SNVs) in the UTRs of 18,593 canonical gene transcripts (see Methods) that would create a new uAUG, yielding 562,196 possible SNVs". "Of these, 15,239 (2.7%) were observed at least once in whole genome sequence data from 15,708 individuals in gnomAD". "We compared the mutability adjusted proportion of singletons (MAPS) score, a measure of the strength of selection acting against a variant class²³, for 14,897 observed high-quality autosomal uAUG-creating SNVs".

** Earlier work (incl. PMID26332131) looking at the conservation of noncoding sequence across genes did find that dosage sensitive genes have a higher conserved UTR. To disentangle the current correlations with pLI and dosage sensitive genes, is there a way to potentially show whether 5' UTR start variant sites are found to be more highly conserved relative to the corresponding gene's overall 5' UTR sequence?*

Response: This is a great question which we took some time to explore. We have now included a new supplementary figure and the following results text "To ensure that the observed increase in MAPS score upstream of pLoF intolerant genes is not purely because the UTRs of these genes are more highly conserved, we compared the conservation of potential uAUG sites with the remainder of the 5'UTR, across all sextiles of LOEUF score. Overall, a significantly higher proportion of possible uAUG-creating bases have phyloP scores >2 (10.3%), when compared to all other 5'UTR bases (8.6%; Fisher's $P < 1 \times 10^{-100}$), with the size of this effect increasing as the corresponding genes become more intolerant to pLoF variants (Supplementary Figure 2)."

** There is a growing catalogue of accessible pediatric disease trio WGS, including from the 100K genomes project and other large-scale developmental disorder collections. Often the de novo significance enrichment identified for these ID/DD genes is driven by LoF mechanisms. As such, is there an opportunity to construct a statistical framework to address whether there is an overall elevated rate in 5' UTR start variants among ID/DD trio cohorts given the total sequence context occupied by all possible such variants (562,196)? Something along these lines could considerably assist in understanding whether there is a significant contribution of this class of variants to overall ID/DD architecture. Perhaps focused on oORF, given it's MAPS score relative to pLOF alleles.*

Response: We made every effort as part of our analyses to identify large disease trio WGS datasets, particularly for ID/DD, which would afford enough power for an analysis such as this. Unfortunately, studies published to date (including the DDD project) use WES data (with some top-up to include targeted non-coding regions that do not include UTRs in the case of DDD). In addition, access to analyse cohorts within the 100K genomes project is restricted to sub-groups with particular disease focuses with long embargo periods on data export and publication after approval of a specific project. We hope to analyse all individuals within this dataset for uORF-perturbing variants in the future (through membership of a cross-cutting analysis domain), however, given these restrictions the ability to do so did not fall within the scope and timeline of the current study.

** For the past year or so, TOPMED has represented a considerably larger resource of WGS than gnomAD. Were TOPMED 5' UTR variants considered?*

Response: Yes, we agree that the TOPMed dataset is a valuable WGS resource and is indeed larger than the current gnomAD release. However, we have limited insight into the sample and variant quality control in TOPMed, and the individual-level data from the project is currently extremely difficult to access for non-consortium members. In addition, our understanding is that related individuals have not been pruned from the public TOPMed summary data (which can distort the site frequency spectrum). We look forward to the full availability of datasets such as TOPMed for similar analyses in the future, however, we believe that repeating our current analysis to include the TOPMed data in its current state is out of scope for this paper.

Reviewer #3 (Remarks to the Author):

I was asked to review and comment on the pathogenicity claims pertaining to neurofibromatosis in this manuscript in light of the claimed 5' UTR variant deleteriousness. Although I read the entire manuscript and supplements, I focus especially on the claims related to neurofibromatosis. Overall, I found the work well-written, measured and plausible.

Neurofibromatosis type 1. There are four NF1 variants originally reported in Evans et al EBioMedicine 2016 and characterized in the current manuscript as uAUG-creating or stop-removing (Figure 3). I was a reviewer on the Evans 2016 paper, and thought it was

a well-done study of NF1 variation. I re-examined the paper and the description of the four 5'-UTR NF1 variants (Table 4 of Evans 2016). There the authors note that non-specific NF1 features segregate with variation in two families; two other families had apparent de novo disease. No other report cataloged in HGMD describes such 5'-UTR variation. In NF1, claims of pathogenicity of 5'-UTR variation (or any non-splice regulatory region) are rare to non-existent. That said, it has been long-recognized that sequencing of NF1 detects >95% deleteriousness variation in people who meet NF1 diagnostic criteria, even when accounting for Legius syndrome. Certainly 5'-UTR variation may explain some of that "missing" <5% variation and this paper is a useful addition to identifying a plausible source of that variation.

Given the assertion that those four NF1 5'-UTR variants are pathogenic, the authors could strengthen their argument by documenting more clinical details from the families/individuals that harbored that variation, with pedigrees. (The original work was done in Manchester by Gareth Evans's group and I see that one of the authors is affiliated with St. Mary's in Manchester.) The authors should also verify and assert (if true) that no other NF1 variation (including deletion testing) was detected in those individuals. If possible, documentation of negative SPRED1 sequencing would be helpful, especially if the NF1 patients had pigmentary findings. If available, tumors with the putative 5'-UTR plus a somatic second hit would be additional convincing evidence.

Response: We have now included pedigrees for the individuals with available family data in Supplementary Figure 3a. We have also clarified the lack of other NF1 or SPRED1 variants in these individuals by including the following in the main text "These variants were found in six unrelated probands, all of whom were negative for coding variants in both *NF1* and *SPRED1*." Unfortunately, tumour data was not available to confirm a somatic second hit in these individuals.

Neurofibromatosis type 2. I have similar comments for NF2, however, the sequenced cohort has not been published. I would prefer a much more detailed accounting of the NF2 phenotype, with pedigrees. If available, tumors with the putative 5'-UTR plus a somatic second hit would be additional convincing evidence.

Response: We have included more detailed phenotypic information on the two *NF2* probands and their affected family members in Supplementary Table 4 and more details on how these patients were diagnosed in the methods section (specifically we have added the sentence "All individuals fulfilled both Manchester and NIH criteria for diagnosis."). In addition, we have included pedigrees for these individuals in Supplementary Figure 3b. As with the *NF1* individuals, tumour data was not available to confirm a somatic second hit in these patients.

More minor comments:

1) Prefer either "neurofibromatosis type 1" instead of just "neurofibromatosis" when referring to the consequence of pathogenic variation in NF1.

Response: This has been corrected for all instances in the text.

2) In Supplemental Tables 1 and 2, please add HGMD accession numbers for given variants. Although I was able to locate the referenced variation, it would have been much easier to do so if given the proper HGMD accession number. Even better would be the PubMed ID(s) for publications reporting particular variation. This way readers can quickly judge for themselves the phenotypic consequence of the uAUG-creating or stop-removing variants

Response: This is a great point. We have now included PubMed IDs in supplementary tables 1 and 2.

Reviewers' Comments:

Reviewer #2:

Remarks to the Author:

Thank you for addressing my comments. No additional comments.

Reviewer #3:

Remarks to the Author:

The authors have responded to my comments to my satisfaction. This is a very nice paper and I look forward to looking for uORFs in my own analyses.